# Atypical Presentation of Methicillin-Susceptible *Staphylococcus aureus* Infection in a Dengue-Positive Patient: A Case Report with Virulence Genes Analysis

**DOI:** 10.3390/pathogens9030190

**Published:** 2020-03-05

**Authors:** Soo Tein Ngoi, Yee Wan Lee, Wen Kiong Niek, Foong Kee Kan, Sazaly AbuBakar, Sasheela Sri La Sri Ponnampalavanar, Nuryana Idris, Cindy Shuan Ju Teh

**Affiliations:** 1Department of Medical Microbiology, Faculty of Medicine, University of Malaya, 50603 Kuala Lumpur, Malaysia; wenkiong308@gmail.com (W.K.N.); sazaly@um.edu.my (S.A.); nuryana@ummc.edu.my (N.I.); 2Department of Medicine, Faculty of Medicine, University of Malaya, 50603 Kuala Lumpur, Malaysia; ywlee@ummc.edu.my (Y.W.L.); sheela@ummc.edu.my (S.S.L.S.P.); 3Infectious Disease Physician, Gleneagles Hospital Medini Johor, 79250 Iskandar Puteri Johor, Malaysia; kanfoongkee@gmail.com; 4Tropical Infectious Diseases Research & Education Centre (TIDREC), University of Malaya, 50603 Kuala Lumpur, Malaysia

**Keywords:** bacteremia, dengue, infectious disease, MSSA, *Staphylococcus aureus*

## Abstract

Concurrent bacteraemia in patients with dengue fever is rarely reported. We report a case of a patient who initially presented with symptoms typical of dengue fever but later succumbed to septic shock caused by hypervirulent methicillin-susceptible *Staphylococcus aureus* (MSSA). A 50-year-old female patient with hypertension and diabetes mellitus presented with typical symptoms of dengue fever. Upon investigation, the patient reported having prolonged fever for four days prior to hospitalization. Within 24 hours post-admission, the patient developed pneumonia and refractory shock, and ultimately succumbed to multiple-organs failure. Microbiological examination of the blood culture retrieved a pan susceptible MSSA strain. Genomic sequence analyses of the MSSA strain identified genes encoding staphylococcal superantigens (enterotoxin staphylococcal enterotoxin C 3 (SEC3) and enterotoxin-like staphylococcal enterotoxins-like toxin L (SElL)) that have been associated with toxic shock syndrome in human hosts. Genes encoding important toxins (Panton-Valentine leukocidins, alpha-haemolysin, protein A) involved in the development of staphylococcal pneumonia were also present in the MSSA genome. *Staphylococcus aureus* co-infections in dengue are uncommon but could be exceptionally fatal if caused by a toxin-producing strain. Clinicians should be aware of the risks and signs of sepsis in dengue fever, thus allowing early diagnosis and starting of antibiotic treatment in time to lower the mortality and morbidity rates.

## 1. Introduction

*Staphylococcus aureus* (*S. aureus*) infection manifests in a wide range of clinical symptoms, including but not limited to bacteraemia, infective endocarditis, infections of the skin, soft tissue, bones, and joints; pneumonia, device-related infections, meningitis, toxic shock syndrome, and urinary tract infections [1]. One of the most frequently encountered infections caused by *S. aureus* is bacteraemia, which is often associated with significant mortality once manifested in patients [2]. Multiple factors influence *S. aureus* bacteraemia mortality, including host factors, pathogen–host interactions, and pathogen-specific factors [3]. *S. aureus* is known for its ability to secrete multiple toxins for enhanced pathogenicity in human hosts, which often results in more severe clinical outcomes, such as toxic shock syndrome and sepsis death [4]. Staphylococcal enterotoxins (SEs) are the most prominent exotoxins produced by *S. aureus* during infections in humans. The superantigenic SEs over-stimulate T-lymphocytes and subsequently induce a cytokines storm in the host’s immune response, which eventually leads to severe hypotension associated with toxic shock syndrome [4]. 

Dengue fever is one of the vector-borne diseases of great concern to public health in Malaysia. The Ministry of Health (MOH) Malaysia has documented high incidence rates of dengue fever since 2013, averaging more than 300 cases per 100,000 population on a yearly basis [5]. The endemicity of dengue virus infections in this region is attributable to the tropical climate and suboptimal environmental cleanliness due to dense populations [6,7]. While in itself is a potentially life-threatening disease, dengue patients are often predisposed to co-infections by bacteria, fungi, protozoa, and other viruses, due to dengue-associated immunosuppression [8,9,10,11]. Concurrent bacterial infections in dengue patients are rare but the number of cases is increasingly being reported within the past decade [11]. Most often, dengue with multiple infections has resulted in more severe disease manifestations and increased patient mortality [8,12]. It is suggested that more than half of the critically-ill dengue patients admitted to the intensive care units had concomitant bacterial infections [12]. 

Bacteria in the *Enterobacteriaceae* family, such as *Escherichia coli* and *Salmonella* spp., have been identified and remain the major causative agents recovered from dengue infections [13,14]. However, reports of staphylococcal co-infections in dengue patients have been increasingly documented in recent years, albeit at very low frequencies [14,15,16,17,18]. Both the methicillin-resistant and susceptible strains of *S. aureus* have been identified with the latter being the predominant phenotype [14,18,19]. Although methicillin-susceptible *S. aureus* (MSSA) appears to be more common than the methicillin-resistant *S. aureus* (MRSA), the specific cause of its predominance remains unclear. We report herein a fatal case of a community-acquired staphylococcal co-infection in a dengue patient, caused by a strain of MSSA designated as hypervirulent due to the presence of multiple virulence determinants in the genome of the bacterium. 

## 2. Case Presentation

### 2.1. Case History

A 50-year-old lady, Madam N, was admitted to a hospital in the Segamat district of Johor, Malaysia with acute fever for four days with no diurnal variation or precipitating factors. It was associated with headache, nausea, myalgia, arthralgia, lethargy and reduced oral intake. She had diabetes mellitus and hypertension, but she was not compliant with her treatment and follow up. Her glycaemic control was poor with glycated haemoglobin (HbA1c) of 10.4%. Upon presentation to the emergency department on 15 February 2017, her condition was stable with a blood pressure of 135/75 mmHg and a heart rate of 90 beats per minute. She was febrile at 38.7 °C. Clinically, she had good pulse volume, warm peripheries, and examination of the cardiovascular, respiratory and abdominal system was unremarkable. Further history revealed that she worked as a canteen operator. There was no history of any recent travels, but she stayed in a dengue-endemic area in Malaysia, where dengue fever is a major public health concern. 

Her initial blood tests showed haemoglobin of 10.4 g/dL, white blood cell count 6.4 × 10^9^/L, platelet 102 × 10^9^/L and haematocrit of 31.1%. Her renal function and liver function test did not show any significant abnormality at that time (Table 1). A presumptive diagnosis of dengue fever day 4 of illness was made. A dengue serology rapid test was done which showed dengue NS1 positive, while dengue IgM and IgG were negative by Rapid test. 

As the initial assessment was stable, she was admitted to the general medical ward. She was started on intravenous normal saline hydration of 2 cc/kg/hr. Upon review in the morning, about 8 hours after admission, she was noted to be breathless. Her peripheries were cold, and saturation was only 92% under high flow oxygen of 15 L/min. Her blood pressure was 180/90 mmHg, and she was tachycardic at 148 beats/min. Auscultation of the lungs revealed bilateral crepitations. Arterial blood gas showed severe lactic acidosis with type 1 respiratory failure. She was immediately transferred to the intensive care unit (ICU) and was intubated. She was given a dose of ceftriaxone 2 g to treat for pneumonia. The total fluid balance overnight was positive 1505 cc.

Over the next 12 hours, her condition deteriorated rapidly despite adequate fluid resuscitation. Fluid given was adjusted accordingly to central venous pressure monitoring. She was also given a whole blood transfusion and supported with maximum inotropic support. Antibiotics were changed to meropenem as her condition further deteriorated and doxycycline was added for atypical infections coverage. There was no evidence of any occult bleeding to suggest dengue haemorrhagic fever. She remained persistently lactic acidosis. Unfortunately, the district hospital did not have continuous renal replacement therapy service and she was not stable to be transferred to a tertiary centre. A bedside transthoracic echocardiogram showed a good heart function with an ejection fraction of 65%, no regional wall motion abnormalities and no obvious vegetation. An electrocardiogram showed sinus tachycardia with no ischaemic changes to suggest a cardiac event. 

She developed multi-organ dysfunction and refractory shock, and she succumbed to her condition in less than 24 hours from admission. Subsequently, her blood culture grew MSSA and her C-reactive protein was elevated. Her hepatitis B, hepatitis C and human immunodeficiency virus (HIV) screening was negative. There was no obvious wound or abscess seen on clinical examination. In view of her rapid deterioration and subsequent death due to severe sepsis with concomitant dengue fever, we sent her blood culture sample for further genomic analysis to further study the strain’s virulence and pathogenicity factors. 

### 2.2. Identification of Virulence Determinants in MSSA Genome

The MSSA strain was sent to a research laboratory at the University of Malaya for genomic assessment and was coded as HS-MSSA. The genomic DNA of the HS-MSSA strain was extracted using a commercial DNA extraction kit (Qiagen, Hilden, Germany) and subjected to whole-genome sequencing via the Illumina Miseq platform (GA2x, pipeline version 1.80) by a commercial sequencing vendor. This Whole Genome Shotgun project has been deposited at the National Center for Biology Information (NCBI) GenBank under the accession number VCMW00000000. The version described in this paper is version VCMW01000000. Virulence determinants in the draft genome of HS-MSSA were identified by using the web-based tool VFanalyzer which retrieves virulence gene sequences from the curated database of Virulence Factors for Pathogenic Bacteria (VFDB; http://www.mgc.ac.cn/VFs/) [17]. Virulence factors related to bacterial adherence, enzyme production, host immune evasion, secretion system and toxins production were identified in the genome of the HS-MSSA strain (Table 2). Multiple toxin genes encoding for exfoliative toxin (type A), haemolysins (alpha, beta, gamma, and delta), staphylococcal enterotoxin C (subtype SEC3) and SE-like toxin L (SElL), exotoxins, leukotoxins, and Panton-Valentine leukocidins (PVL) were identified in the genome of HS-MSSA strain. The global accessory gene regulator operon (*agr*ABCD) and the staphylococcal accessory regulator (*sar*) system were also identified in the HS-MSSA genome. 

### 2.3. Antimicrobial Susceptibility Testing 

The Kirby–Bauer disk diffusion method was used to determine the antimicrobial susceptibility profile of the HS-MSSA strain according to the Clinical and Laboratory Standards Institute (CLSI) guideline (2018) [20]. The HS-MSSA strain was found to be pan-susceptible to all antimicrobial agents tested. The growth of the HS-MSSA strain was effectively inhibited by amoxicillin–clavulanate, cefoxitin, ceftriaxone, ciprofloxacin, clindamycin, cloxacillin, erythromycin, fusidic acid, gentamicin, penicillin-G, piperacillin-tazobactam, rifampicin, and sulfamethoxazole–trimethoprim. 

## 3. Discussion

This is the first case report of recovery and identification of a hypervirulent MSSA co-infection in a dengue patient in Malaysia. The bacterial infection was classified as community-acquired due to its rapid onset within 24 hours post-admission. The patient initially presented as dengue fever and not in shock at the time of admission. However, the patient rapidly developed symptoms of pneumonia and refractory shock and succumbed to multiorgan failures despite intensive therapeutic efforts. Isolation of an MSSA strain from the patient’s blood culture confirmed bacteraemia in the patient. Genomic sequence analyses revealed multiple virulence determinants within the genome of HS-MSSA. 

Upon admission, the patient was presumptively diagnosed as dengue fever based on clinical presentations, and subsequent serology, tests showed positive dengue NS1 protein. NS1-based dengue tests are often specific (specificity ranged between 86.1% and 100%), but the sensitivity varies greatly depending on the dengue virus serotypes, duration of illness, and types of infection (primary or secondary dengue) [21]. Although uncommon, a false-positive result could occur in dengue NS1 testing due to cross-reactions with other flaviviruses, such as zika virus [22,23]. False-positive dengue NS1 tests have also been documented in patients with haematological malignancies [24]. Simultaneous testing of NS1 with IgM and IgG antibodies is essential for enhanced accuracy in dengue diagnosis [25,26]. However, dengue-specific IgM and IgG antibodies tested negative in the patient’s serum during the serology testing. Nonetheless, the patient was at day 4 of illness during the time of admission, and very often, the detection of dengue-specific antibodies is only possible after 4 or 5 days of disease onset, with acceptable diagnostic accuracy on the sixth day onwards [27,28]. Unfortunately, the patient had succumbed to multiple organ failure within 24 hours post-admission. Hence, further testing could not be done to confirm dengue virus infection.

*S. aureus* often harbours multiple toxin genes in its genome, and the most prominent toxins are the staphylococcal superantigens (staphylococcal enterotoxins, SEs; SE-like proteins, SEls; toxic shock syndrome toxin 1, TSST-1) [4]. The presence of genes encoding for SEC3 and SElL in the HS-MSSA genome infers its ability to produce these potent immunostimulatory superantigenic toxins. These superantigens are capable of binding directly to the major histocompatibility complex class II molecules on the antigen-presenting cells (APC) outside of the antigen groove, thus bypassing APC processing prior to T cell presentation [4]. This subsequently leads to the activation of an abundance of T cells, causing the massive release of chemokines and pro-inflammatory cytokines that eventually result in lethal toxic shock syndrome. The SEC, together with SEB, is the predominant SE serotype of *S. aureus* that causes non-menstrual toxic shock syndrome. Besides superantigenic activity, SEs and SEls are also pyrogenic, emetic (only for SEs), and capable of inducing lethal hypersensitivity to endotoxins [29]. 

Staphylococcal pneumonia is often associated with the production of PVL, protein A, and alpha-haemolysin by *S. aureus* [30]. The presence of these genes in the genome of HS-MSSA could have explained the manifestation of pneumonia in the patient. A previous study by Labandeira-Rey et al. [31] showed that PVL alone is sufficient to cause pneumonia. Moreover, the expression of PVL could further induce changes in the transcriptional levels of the genes encoding both secreted and cell-associated virulence factors in *S. aureus* [31]. Alpha-haemolysin is capable of causing damage to lung epithelial tissue, and when combined with high levels of PVL, often results in severe necrotizing pneumonia in humans [32]. Protein A, at the same time, induces an inflammatory response of the airway by activating the tumour-necrosis factor-α receptor, TNFR1, hence playing an essential role in the pathogenesis of staphylococcal pneumonia [33]. 

PVL-producing *S. aureus* has been found to cause life-threatening infection and does not respond to antimicrobial treatment despite in vitro susceptibility to the antimicrobial agents, resulting in a clinical condition termed PVL syndrome [34]. PVL syndrome is commonly caused by community-acquired *S. aureus* infection, with a higher frequency of methicillin-susceptible subtypes [34]. A similar observation was made in this clinical case, whereby the patient’s condition did not improve after the administration of ceftriaxone, meropenem, and doxycycline, despite in vitro susceptibility of the HS-MSSA strain to these antimicrobial agents later being verified by laboratory tests. In a previously reported PVL-producing *S. aureus* infection, the patient was eventually treated with a combination of clindamycin and daptomycin [34]. This notion supported the empirical use of clindamycin in patients to prevent possible toxin-mediated sepsis. 

Virulence genes expression in *S. aureus* is mainly regulated by the *agr* system (*agr*ABCD and delta-haemolysin) [35,36]. Together with the staphylococcal accessory regulator (*sar*) system, the *agr* system is found to upregulate the expression of SEB and SEC [4]. Besides that, the role of the *agr* system is also proved to be essential in the pathogenesis of acute lethal staphylococcal pneumonia by regulating the expression of alpha-haemolysin [37]. The pathogenicity of the HS-MSSA strain might be attributed to the concerted efforts of the intact *agr* and *sar* systems in regulating the expression of the array of virulence genes in its genome. Moreover, the hypervirulence of the HS-MSSA strain could explain the predominance of MSSA over MRSA in concurrent bacteraemia in dengue patients. Nonetheless, future studies involving more *S. aureus* isolates from dengue dual infections should be conducted to verify this hypothesis. 

The majority of the prominent toxin-producing genes are found in regions with considerable genetic variations, mostly associated with phage regions within the HS-MSSA genome (Unpublished data). Contig-13, which harbours multiple enterotoxins and exotoxins, was predicted as an incomplete phage region associated with the superantigen-encoding staphylococcal pathogenicity island (SaPI). Contig-4 harbours multiple virulence factors including leukotoxins (PVL and LukED) within a phage region that also encodes for SaPIn3-associated proteins. The leukotoxin LukGH identified in contig-11 was found located next to a phage region. All these evidences support the previous notion that highly mobile genetic elements, such as phage-associated pathogenicity islands, play an important role in the evolution of this hypervirulent *S. aureus* lineage by providing survival advantages to enhance its pathogenicity and host-adaptability [38]. 

Most often, the outcome of a disease does not depend solely on the pathogenicity of the etiologic agent, but host factors play an equally important role. The vulnerability in the host immune defence mechanisms and microbial virulence involves a two-way host-pathogen interaction, causing severe sepsis in the host [39]. In the fatal sepsis reported in this case, the patient’s immune system might have been suppressed by the dengue virus, resulting in host vulnerability to MSSA co-infection. Evidence has shown that in neonates and the elderly, immune cells infected by dengue virus produced fewer cytokines leading to an immunosuppressed state of the hosts [40]. Indeed, dengue patients with concurrent bacteraemia are often older and tended to have prolonged fever [41]. In Malaysia, 90% of fatal cases of dengue haemorrhagic fever occurred in adult females with a median age of 32-years and an average of 4.7 days of illness prior to hospitalization [42]. Old age is the most prominent risk factor associated with dengue mortality in Malaysia [43]. Moreover, underlying comorbidities, particularly hypertension and diabetes mellitus, are commonly associated with more severe dengue cases and greater mortality rates [8,12,42]. The combined effect of old age (50-year-old), prolonged duration of fever (4 days), comorbid chronic illnesses (hypertension and diabetes mellitus), and concurrent bacteraemia by an enterotoxin-producing strain of MSSA, might have caused the rapid deterioration and eventually the death of the dengue patient in this case. 

To date, there are still limited effective emergency therapeutic options for established refractory shock. Short-term mortality occurs in 50% of critically ill patients who developed refractory shock [44]. Therefore, early and aggressive interventions should be implemented before refractory shock develops. Nandhabalan and colleagues who are experienced in toxin-mediated sepsis have recommended the use of adjunctive antimicrobial therapy, where clindamycin is administered empirically in addition to broad-spectrum antibiotics until microbiological analyses have proven the absence of toxin-producing pathogens, or until organs’ dysfunction are stabilised [45]. Clindamycin functions to inhibit bacterial protein synthesis and most importantly, prevents the production of superantigens and is clinically proven to be effective in the treatment of toxic shock syndrome [46]. Nonetheless, increasing clindamycin resistance has been observed among *S. aureus* populations during recent years [47]. Therefore, more regional susceptibility data should be collected, and clinicians should be aware of antimicrobial resistance patterns of the MSSA population when choosing empirical regimens. 

This case report documents the first virulence analysis of an MSSA strain isolated in a fatal case of dengue dual infections in Malaysia. The presence of multiple virulence factors, especially the superantigens SEC3 and SElL, together with the regulatory genes, might have contributed to the hypervirulence of the HS-MSSA strain. Although concurrent bacteraemia in dengue patients remains rarely reported, especially one caused by *S. aureus*, the greater mortality risk of such dual infections should not be overlooked. Early diagnosis and interventions are essential to prevent unfavourable clinical outcomes in patients. Upon examination of patients presenting symptoms of dengue fever, the attending physicians should also consider the possibility of false-positive serology test results and bacteraemia. Delayed diagnosis can prove fatal especially when patients are infected by toxin-producing bacterial pathogens, where bacteraemia can rapidly develop into refractory septic shock and death may occur within a short time. Further, the lack of reporting and attention for concurrent bacteraemia in dengue patients in Malaysia necessitates increasing efforts in surveillance. The actual prevalence of co-infections and the characteristics of the etiologic agents may help medical practitioners in making appropriate medical decisions and treatment regimens to prevent dengue mortality. 

## Figures and Tables

**Table 1 pathogens-09-00190-t001:** Blood investigation results ^1^.

Date	15/2	15/2	15/2	16/2	16/2	16/2	16/2
Hb (g/dL)	10.4	14.0	13.1	14.1	14.4	14.1	15.4
WBC (×10^9^/L)	6.4	8.1	7.9	2.2	2.7	1.3	2.2
HCT L/L (%)	31.1	40.8	39	42.7	44.2	42.3	46.9
Platelet (×10^9^/L)	102	126	109	95	121	90	68
Lactate (mmol/L)				6.0	4.8	7.2	9.8
Urea (mmol/L)	4.3			3.4	5.2		
Sodium (mmol/L)	125			127	128		
Potassium (mmol/L)	3.4			3.1	3.2		
Creatinine (µmol/L)	71			62	101		
Bilirubin (µmol/L)	9.1			11.4	13.7		
Albumin (g/L)	38.3			30.9	24.9		
ALP (U/L)	84			68	60		
ALT (U/L)	47.3			41.4	44.2		
AST (U/L)	84.5			165.9			
CK (U/L)	157			2735			
LDH (U/L)	247			406			
CRP (mg/L)				301.21			
ESR (mm/h)				15			
ABG							
pH				7.218	7.228		7.217
pCO_2_ (mmHg)				49.2	42.5		28.8
pO_2_ (mmHg)				46.8	105		101
HCO_3_ (mmol/L)				17.2	16.7		13.2

^1^ Hb: Haemoglobin; WBC: White blood cell; HCT: Haematocrit; ALP: Alkaline phosphatase; ALT: Alanine transaminase; AST: Aspartate transaminase; CK: Creatine kinase; LDH: Lactate dehydrogenase; CRP: C-reactive protein; ESR: Erythrocyte sedimentation rate; ABG: Arterial blood gas; pCO_2_: Partial pressure of carbon dioxide; pO_2_: Partial pressure of oxygen; HCO_3_: Bicarbonate.

**Table 2 pathogens-09-00190-t002:** Virulence genes identified in HS-MSSA genome ^1^.

Virulence Factor	Gene	Contig	Sequence Region (bp)
**Adherence**			
Autolysin	*atl*	2	172,186–175,965
Cell wall-associated fibronectin-binding protein	*ebh*	15	17,112–29,540
Collagen adhesion	*cna*	19	142,053–142,376
Elastin binding protein	*ebp*	15	78,470–79,936
Extracellular adherence protein/MHC analogous protein	*eap/map*	11	50,777–52,522
Fibrinogen binding proteins	*efb*	12	69,305–69,802
Fibronectin binding proteins	*fnbA*	6	7942–11,022
Fibronectin binding proteins	*fnbB*	6	4386–7193
Intercellular adhesin	*icaA*	19	115,805–117,004
Intercellular adhesin	*icaB*	19	117,249–118,142
Intercellular adhesin	*icaC*	19	118,129–119,181
Intercellular adhesin	*icaR*	19	115,042–115,602
Staphylococcal protein A	*spa*	10	91,167–92,729
**Enzyme**			
Cysteine protease	*sspB*	2	166,228–167,361
Cysteine protease	*sspC*	2	165,861–166,190
Hyaluronate lyase	*hysA*	21	38,536–40,908
Lipase	*geh*	32	23,616–25,688
Lipase	*lip*	19	119,572–121,617
Serine V8 protease	*sspA*	2	167,491–168,519
Serine protease	*splA*	4	383,603–384,310
Serine protease	*splB*	4	381,976–382,698
		4	382,756–383,478
Serine protease	*splC*	4	381,199–381,918
Serine protease	*splD*	4	378,589–379,308
		4	380,362–381,078
Serine protease	*splE*	4	377,715–378,431
		4	379476–380192
Serine protease	*splF*	4	376,845–377,564
Staphylocoagulase	*coa*	18	71,941–73,923
Thermonuclease	*nuc*	2	5253–5807
		27	92,177–92,749
Zinc metalloproteinase aureolysin	*aur*	19	67,567–69,096
**Immune evasion**			
Adenosine synthase A	*adsA*	10	54,213–56,555
Capsule	*--*	36	12,417–13,085
		36	13,101–13,787
		36	13,790–14,554
		36	14,763–16,397
		36	16,387–17,415
		36	17,422–18,537
		36	18,541–19,665
		36	19,668–20,747
		36	20,740–22,134
		36	22,131–22,688
		36	22,697–23,935
		36	25,185–25,742
		36	25,742–26,629
		36	26,683–27,945
		36	28,034–29,167
Staphylococcal complement inhibitor	*scn*	11	54,281–54,631
Staphylococcal binder of immunoglobulin	*sbi*	5	117,022–118,335
**Secretion system**			
Type VII secretion system	*esaA*	35	15,975–19,004
Type VII secretion system	*esaB*	35	19,434–19,676
Type VII secretion system	*esaG*	35	30,637–31,077
		35	31,088–31,588
		35	31,599–32,099
		35	32,110–32,610
		35	32,621–33,112
		35	33,739–34,239
		35	34,250–34,741
		32	2354–2854
		32	2865–3365
		32	3376–3864
Type VII secretion system	*essA*	35	19,004–19,462
Type VII secretion system	*essB*	35	19,689–21,023
Type VII secretion system	*essC*	35	21,084–25,493
Type VII secretion system	*esxA*	35	15,599–15,892
**Toxin**			
Alpha hemolysin	*hly/hla*	12	72,363–73,331
Beta hemolysin	*hlb*	11	53,721–53,921
Delta hemolysin	*hld*	38	2974–3108
Enterotoxin C	*sec*	13	58,618–59,418
Enterotoxin-like L	*sell*	13	59,585–60,307
Exfoliative toxin type A	*eta*	3	7742–8689
Exotoxin	*set15*	14	2106–2801
Exotoxin	*set16*	13	65,639–66,319
Exotoxin	*set18*	13	67,591–68,661
Exotoxin	*set21*	13	71,399–72,094
Exotoxin	*set22*	13	72,516–73,211
Exotoxin	*set24*	13	74,632–75,330
Exotoxin	*set25*	13	75,617–76,378
Exotoxin	*set33*	13	69,011–69,901
Exotoxin	*set34*	13	70,250–70,969
Exotoxin	*set37*	13	73,548–74,252
Exotoxin	*set7*	13	66,545–67,300
Gamma hemolysin	*hlgA*	5	118,835–119,800
Gamma hemolysin	*hlgB*	5	121,316–122,293
Gamma hemolysin	*hlgC*	5	120,367–121,314
Leukocidin subunit G	*lukG*	11	10,300–11,316
Leukocidin subunit H	*lukH*	11	9223–10,278
Leukotoxin D	*lukD*	4	388,427–389,410
Leukotoxin E	*lukE*	4	389,412–390,346
Panton-Valentine leukocidin	*lukF-PV*	4	259–1236
Panton-Valentine leukocidin	*lukS-PV*	4	1238–2185

^1^ Details of the predicted virulence genes in the HS-MSSA genome are provided in the Appendix A.

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
