# Peer review of "Atypical Presentation of Methicillin-Susceptible *Staphylococcus aureus* Infection in a Dengue-Positive Patient: A Case Report with Virulence Genes Analysis"

_pathogens, 2020, doi:10.3390/pathogens9030190_

Round 1

Reviewer 1 Report

In this work titled Atypical presentation of methicillin-susceptible Staphylococcus aureus infection in a dengue-positive patient: A case report with virulence analysis Ngoi et al present a case report of a 50 years-old woman admitted with dengue fever symptoms who developed pneumonia, refractory shock and deceased by multiple organ failure in less than 24 hours post admission. Microbiologic culture of blood sample retrieved Staphylococcus aureus and whole genome sequencing showed the presence of expected staphylococcal virulence factors.

Comments:

  • Acronyms should be explained the first time they appear in the main text, e.g. Panton-Valentine Leukocidin: PVL, and so on.
  • At the time of this review, although the authors claim that they have deposited the project into GenBank, access to the sequence is not possible. It would be necessary to include a new column with the gene accession number in table 2 so that readers can access genetic information more easily.
  • Since toxins and virulence genes are mostly encoded by mobile elements, why the authors did not include information regarding phages and pathogenicity islands? How many phages does this strain harbor? How many pathogenicity islands? Which toxins and virulence genes are encoded by which MGEs? Is SCC present but lacks the mecA gene? Is ACME present?
  • A figure with a circular genome map should be included, giving answer to all questions in previous point.

Reviewer 2 Report

The study by Ngoi et al. presented a case report about MSSA infection in a dengue-positive patient. Authors precisely analyzed patient's history as well as general blood results. Blood culture gave MSSA-positive result which prompted authors to determine the whole susceptibility profile of this pathogen. Overall, the data are of significance to the field ofdengue fever pathogenesis and co-infections. Several concerns are outlined below.

Line 73: consider 'was admitted' instead of 'presented'

Line 122: a producer od DNA extraction kit is missing

- consider the percentage entry without spaces

Reviewer 3 Report

The authors present a severa case of pneumonia and sepsis with unfavorable evolution.

This is a well presented case, in chronological order, easy to follow. The laboratory part is as well properly presented.

Author Response

We thank the reviewer for the positive comments. 

Round 2

Reviewer 1 Report

To me there is not conflict by showing which mobile genetic elements are present in this strain and which mobile elements encode each MGE.

Why authors don't mention nor reference their somewhere-else-submitted manuscript?

The information table 2 is showing now regarding contigs and seq position will be useless and confusing to the scientific community once the sequence is finally assembled, deposited and publicly accessible. 
